# Global distribution and variability of subsurface chlorophyll a concentrations

Sayaka Yasunaka1, Tsuneo Ono2, Kosei Sasaoka1, Kanako Sato1

1Japan Agency for Marine-Earth Science and Technology, Yokosuka, 237-0061, Japan

2Japan Fisheries Research and Education Agency, Yokohama, 236-8648, Japan

Correspondence to: Sayaka Yasunaka (yasunaka@jamstec.go.jp)

**Abstract.** Chlorophyll a (Chl-a) often exhibits a maximum concentration in the subsurface layer rather that at the surface. The depth of the Chl-a maximum primarily depends on the balance between light penetration from the surface and the nutrient supply from the deep ocean. However, a global map of subsurface Chl-a concentrations based on observations has not been presented yet. In this study, we integrate Chl-a concentration data from recent biogeochemical floats as well as from historical ship-based and other observations, and present global maps of subsurface Chl-a concentrations with related variables. The subsurface Chl-a maximum was observed globally throughout the oceans; at depths greater than 80m in the subtropics and tropics (30°S to 30°N), in the 40–80 m depth range in the tropics, in the Southern Ocean (south of 40°S), and in the mid-latitudes (30–40°N/S) in the North Pacific, and at depths of less than 40m in the northern subarctic (north of 40°N). The observed maxima all lie below the mixed layer depth for the entire year in the subtropics and tropics, and during summer in the mid-latitudes and the northern subarctic. The depths of the subsurface Chl-a maxima are greater than those of the photosynthetically active layer in the subtropics but shallower in the tropics and midlatitudes. In the subtropics, a seasonal increase in oxygen below the mixed layer implies substantial new biological production, which corresponds to 10% of the net primary production in that region. During El Niño, subsurface Chl-a concentrations are higher in the middle and eastern equatorial Pacific but lower to the west in comparison with La Niña, a pattern which is opposite to that on the surface. The spatiotemporal variability of the Chl-a concentrations described here has implications to not only for the biogeochemical cyclying in the ocean but also for understanding the thermal structure and dynamics of the ocean via absorption of shortwave radiation.

# 1 Introduction

Chlorophyll a (Chl-a) concentrations in the ocean often exhibit a maximum value not at the surface, but rather in the subsurface layer. The subsurface Chl-a maximum is a widespread and common feature in various oceans in the tropics, subtropics, and subarctic (Anderson, 1969; Saijo et al., 1969; Furuya, 1990; Bhattathiri et al., 1996; Mann and Lazier, 1996). The same finding was also recently reported in the Arctic Ocean and the Southern Ocean (Ardyna et al., 2013; Baldry et al., 2020). Substantial primary production is also observed in the subsurface, although the relationship between Chl-a concentrations, biological biomass, and primary production is not simple, and the subsurface Chl-a maximum often

represents a photoacclimation response in phytoplankton (Kitchen and Zaneveld, 1990; Campbell and Vaulot, 1993; Goldman, 1988; Fennel and Boss, 2003; Matsumoto and Furuya, 2011; Cornec et al., 2021). The depth of the Chl-a maximum primarily depends on the balance between light penetration from the surface and nutrient supply from the deep ocean (Cullen, 2015), but also partially depends on light-dependent grazing by zooplankton near the surface (Moeller et al., 2019).

Chl-a concentrations have physical effects in the ocean because Chl-a absorbs shortwave radiation, which leads to ocean warming followed by a modified thermal structure and ocean dynamics (Lewis et al., 1990; Siegel et al., 1995). Modeling studies have indicated that the interannual variation in subsurface Chl-a concentrations is an important parameter for accurate El Niño simulations (Jochum et al., 2010; Kang et al. 2017).

Using satellite-retrieved ocean-surface Chl-a concentrations, many studies have described the basin-scale spatial distribution and temporal variation of surface Chl-a (e.g. Dunstan et al., 2018; Lin et al., 2014; Sasaoka et al., 2011); however, those features have not been described in detail for subsurface Chl-a. Although several studies have parameterized the vertical profile of Chl-a concentrations and reproduced subsurface Chl-a maxima (Ardyna et al., 2013; Uitz et al., 2006), their main purpose was to estimate depth-integrated Chl-a and primary production. Recently, Biogeochemical Argo floats with Chl-a sensors have revealed occurrence of the subsurface Chl-a maxima and their relationship to phytoplankton biomass across the world's oceans (Cornec et al., 2021). However, subsurface Chl-a data derived from a single data source does not have sufficient coverage to illustrate the wider picture; therefore a global map of observed subsurface Chl-a concentration is still needed. Global maps of subsurface Chl-a maxima have thus far been based on statistical estimates or numerical models (Mignot et al., 2014; Masuda et al., 2021), and only surface Chl-a concentration have been used to validate numerical models (e.g. Séférian et al. 2020).

Here, we synthesize data on Chl-a concentrations data from recent biogeochemical floats as well as from historical ship-based and other observations, and present global maps of subsurface Chl-a concentrations. We then present seasonal and interannual variability of the subsurface Chl-a concentrations in relation to other variables in the world's oceans.

## 2 Data

Chl-a measurements were extracted from the World Ocean Database 2018 (WOD2018; Boyer et al., 2018; https://www.nodc.noaa.gov/OC5/WOD/pr_wod.html) and the Global Ocean Data Analysis Project version 2.2019 Release (GLODAPv2.2019; Olsen et al., 2019; https://www.glodap.info/). These measurements were taken from bottle samples, CTD fluorescence, underway CTD fluorescence, profiling floats, gliders, and drifting buoys. Data from the Biogeochemical Argo floats are included in WOD2018 under the category of measurements from profiling floats. The total number of Chl-a measurements is 114,107,161 from 737,469 profiles measured between 1932 and 2020 in the upper 300 m (Table 1). The data are globally distributed across the oceans (Figure 1). Most of the data for in the open ocean are from bottle samples, CTD fluorescence, and profiling floats (Figure S1).

Chl-a data often include several high values (>3 mg/m³) that represent erroneous data or data that reflect in short-term and

small-scale extreme conditions (Figure S2a). We conducted quality control efforts to reduce the effect of such data, as they might otherwise detract from our purpose of determining large-scale distribution patterns in the open ocean. Although different types of data errors are present among the different data sources, statistical quality control measures using a limited number of data did not work effectively, so we instead uniformly treated the data from all data sources. The quality control measures used are as follows:

1. We binned the Chl-a measurements from each profile into depths of 5 m (0–5 m), 10 m (5–15 m), 20 m (15– 25 m), 30 m (25–40 m), 50 m (40–62.5 m), 75 m (62.5–87.5 m), 100 m (87.5–112.5 m), 125 m (112.5–137.5 m), 150 m (137.5–175 m), and 200 m (175–250 m).

2. We calculated the long-term mean and its standard deviation within ranges of ±5° latitude, ±10° longitude, and ±1 month (regardless of the year) for each 1° × 1° × 1-month grid cell at each depth.

3. We flagged data that differed by more than three standard deviations from the long-term mean in each grid cell at each depth.

4. We eliminated profiles in which more than half of all data had been flagged in Step 3.

5. We eliminated profiles in which more than half of all profiles within ±10° of the latitude and longitude, and ±1 month had been eliminated in Step 4.

This procedure identified approximately 1% of the measurements as belonging to erroneous or extreme profiles (Table 1). Data with high values were extensively eliminated, and 85% of the eliminated data have values of > 3 mg/m³ (Figure S2b). The eliminated data were mostly located in coastal regions and partly scattered in the open oceans (not shown here). The ratio of eliminated data is slightly larger in data from underway CTD fluorescence and bottle samples likely because they included more uncalibrated and historical data (Table 1).

For the remaining data after quality control, we calculated monthly means of Chl-a concentrations in 1° × 1° grid cells at each depth. We also calculated the depth of the Chl-a maximum in each individual profile that included data from more than five different depths, and then binned them into 1° × 1° × 1-month grid cells. The average sampling depth interval around the Chl-a maximum is 7 m (3 m in CTD fluorescence and profiling floats, and 16 m in bottle samples).

Satellite-derived surface Chl-a concentrations, euphotic layer depths (1% light level; Z_eu), and surface values of

photosynthetically available radiation (PAR) with a 1° × 1° monthly resolution since September 1997 were obtained from the GlobColour project website (GlobColour_R2018; http://hermes.acri.fr/index.php; Frouin et al., 2003; Maritorena et al., 2010; Morel et al., 2007). We estimated the PAR within the water column using the empirical relationship between Z_eu and the surface PAR as Ito et al. (2015) did. The PAR at a depth of z was calculated as

$$PAR(z) = 0.98 \times PAR(0) \times \exp(-kz), \quad (1)$$

where k is the light attenuation coefficient within the water column derived from

$$k = -\log(0.01)/Z\_eu. \quad (2)$$

The coefficient 0.98 in (1) is the transmission rate into the ocean used by Boss and Behrenfeld (2010). We then defined photosynthetically active layer as having > 0.415 einstein/m$^2$/day of PAR as per Boss and Behrenfeld (2010).

Monthly fields of net primary production (NPP) were obtained from the Ocean Productivity website (http://www.science.oregonstate.edu/ocean.productivity/index.php) with a spatial resolution of 1/6° × 1/6°, which were calculated using the vertically generalized production model of Behrenfeld and Falkowski (1997). We used climatological means of oxygen concentrations, oxygen saturation ratios, and nitrate concentrations in the World Ocean Atlas 2018 (WOA2018; https://www.nodc.noaa.gov/OC5/woa18/; Garcia et al., 2018a, 2018b). We also used climatological means of mixed layer depths (the depth at which sigma-θ changes by 0.125 compared to that at the surface) produced by JAMSTEC (MILA_GPV; http://www.jamstec.go.jp/ARGO/argo_web/argo/?page_id=223andlang=en; Hosoda et al. 2010; missing data were interpolated using data from the surrounding grids). Using a mixed layer depth defined by a sigma-θ change of 0.03 produced very similar results. We show the results using the mixed layer depth with a change from the surface sigma-θ of 0.125 here.

We calculated the Niño 3.4 index (sea surface temperatures over 5°N to 5°S, 170–120°W) as an indicator of El Niño and La Niña using the Hadley Centre Sea Ice and Sea Surface Temperature data set (HadISST; https://www.metoffice.g.,ov.uk/hadobs/hadisst/; Rayner et al., 2003). El Niño or La Niña were originally defined by periods when the Niño 3.4 index exceeds ±0.4°C for 6 months or longer (Trenberth, 1997). Here, El Niño or La Niña is taken to refer to all positive or negative Niño 3.4 indices, respectively, because the amount of subsurface Chl-a data would otherwise be limited.

## 3 Results

### 3.1 Climatological mean state

Figures 2a and 3 show the depths of the Chl-a maxima and cross-sections of Chl-a concentrations from the quality-controlled data, respectively. We selected the central latitudinal and longitudinal bands of the subtropics, the tropics, the Indian Ocean, the Pacific Ocean, and the Atlantic Ocean in Figure 3. The Chl-a concentrations exhibit subsurface maxima across the world's oceans, at depths below 80 m in the subtropics, at 40–80 m depth in the tropics, the Southern Ocean, and the mid-latitudes in the North Pacific, and at depths above 40 m in the northern subarctic. The subsurface maximum is deeper than 120 m in the central subtropics and reaches 150 m at approximately 25°S, 100°W in the South Pacific. The subsurface maxima in the subtropics and the tropics were deeper than the mixed layer (Figures 4a and 5). In the Southern Ocean and the northern North Atlantic, several patches of subsurface maxima were observed at depths greater than 80 m, while these depths are generally shallower than the mixed layer depth (Figures 2a, 4a, and 5d–f). The ranges of Chl-a concentration at the subsurface maxima are 0.1–0.2 mg/m$^3$ in the subtropics, 0.2–0.5 mg/m$^3$ in the tropics, and >0.5 mg/m$^3$ in the subarctic (Figures 3 and 6a).

The long-term mean of the photosynthetically active layer depth is greater than 80 m in the subtropics, and deeper than 40 m in other regions (Figure 2b). It is also greater than the mixed layer depth in the subtropics, the Arctic Ocean, and the tropics

but shallower in the northern subarctic and the Southern Ocean (Figures 2b and 5). The PAR values at the subsurface Chl-a maxima are generally stronger than 1 einstein/m²/day in the subarctic and in the tropics (Figures 6b). The spatial distribution of the photosynthetically active layer depth is similar to that of the subsurface Chl-a maximum depth (Figures 2a and 2b), but substantial differences were observed (Figure 4b). For example, the subsurface Chl-a maximum is deeper than the photosynthetically active layer in the subtropics and at 40–60°S in the Indian sector, but is shallower than the

photosynthetically active layer depth in other regions.

Nitrate concentrations in the surface layer were less than 1 µmol/kg in places with a deep subsurface Chl-a maximum (Figures 2c and 5). In these areas, the nitrate concentrations at the depth of the subsurface Chl-a maximum were also low, while nitrate concentrations are greater than 5 µmol/kg in the subarctic and eastern tropics but less than 1 µmol/kg in the subtropics and the Arctic Ocean (Figure 6c).

The subtropics are oversaturated with oxygen down to depths of at least 40 m and deeper than 80m in some places (Figures 2d and 5). The lower limit of oxygen oversaturation in the subtropics mostly occurs below the mixed layer and above the subsurface Chl-a maximum (Figures 2d and 4c).

## 3.2 Seasonal variation

The subsurface Chl-a maximum in the subtropics was observed to occur below the mixed layer in both winter and summer

(Figures 7a and 8). The photosynthetically active layer is deeper than the mixed layer during both seasons between 20°N and 30°S in the Indian Ocean and the western Pacific, and between 20°N and 20°S in the Atlantic Ocean and the eastern Pacific, and in the whole summer hemisphere except for 40–60°S (Figures 7b and 8). The photosynthetically active layer is deeper in summer than in winter in nearly all areas except for the North Atlantic (Figure 7e). This summer deepening is more than 15 m in the 20–40° latitudinal bands in both hemispheres (Figures 7e and 8f). The seasonal differences in the Chl-a maximum

depths exhibit the same tendencies as the photosynthetically active layer (Figure 7d). Below the mixed layer, oxygen is oversaturated north of 10°N and at 10–45°S in summer (Figures 7c, 8b, 8d, and 8e). The subsurface oxygen concentrations are higher in summer than in winter at latitudes of approximately 15–40° (Figures 7f, 8c, and 8f).

The subsurface Chl-a maximum deeper than the mixed layer was observed to be stable in the subtropics and tropics (30°S to 30°N), was seen only in summer at midlatitudes (30° to 40°), and occurred rarely at 45–60°S in the Southern Ocean and in

the northern North Atlantic (north of 45°N) (Figures 9a and 10). The percentage of months with a deeper photosynthetically active layer than the mixed layer has a similar pattern to the percentage of months deeper the subsurface Chl-a maximum than the mixed layer, except in the northern North Atlantic (Figure 9b). At the midlatitudes, the Chl-a concentrations were high (>0.3 mg/m³) throughout the entire mixed layer during winter, and remained high at the subsurface but became low on the surface in summer, when the mixed layer becomes shallow (Figure 10c). At 45–60°S in the Southern Ocean and the

northern North Atlantic, the Chl-a concentrations in the mixed layer were high in summer and low in winter, while the

subsurface Chl-a maximum occurred around the mixed layer depth in summer and within the mixed layer in winter (Figures 10h and 10i). In the subarctic North Pacific, the seasonal cycle of the Chl-a concentrations is similar to that in the subarctic North Atlantic, although the surface Chl-a concentrations is relatively low and the subsurface Chl-a maximum appears at a depth below the mixed layer in midsummer (Figure 10g).

**3.3 El Niño–Southern Oscillation (ENSO)-related variation**

A subsurface Chl-a maximum along the equator can be seen in the photosynthetically active layer during El Niño and La Niña (Figures 11a and 11b). Below 40 m which is approximately the mixed layer depth in that area, the Chl-a concentration is lower west of 160°E during El Niño but higher east of 170°W (Figure 11c). Although the difference in the Chl-a concentrations between El Niño and La Niña is noisy above 40 m, most of the significant difference is negative east of 150°E (Figure 11c). Satellite-derived surface Chl-a concentrations show lower values during El Niño than those during La Niña east of 150°E (Figure 12a). Surface PAR is lower (higher) to the east (west) of 150°E during El Niño (Figure 12b), while the photosynthetically active layer deepens by several meters during El Niño east of 150°E (Figure 12c). The lower and higher subsurface Chl-a concentrations to the west and east of 160°E during El Niño occurs mainly at the depths of 60–120 m, which corresponds well with the base of the photosynthetically active layer (Figures 11c and 12c).

**4 Discussion**

The spatial distribution of the subsurface Chl-a maximum in the subtropics seen in this study (Figures 2a and 3) corresponds to the bowl-shaped thermocline structure of the subtropical gyre (Pedlosky, 1990). The deep thermocline in the central subtropics inhibits nutrient supply to the surface, causing low nutrient levels and Chl-a concentrations there.

Uitz et al. (2006) explained that a deeper subsurface Chl-a maximum corresponds to lower surface Chl-a concentrations because light proceeds downward until it is absorbed by chlorophyll. Our study demonstrates that the depth of the subsurface Chl-a maximum is roughly consistent with the depth of the photosynthetically active layer throughout the world's oceans (Figures 2a and 2b). However, Chl-a concentrations in the central subtropics exhibit a maximum below the photosynthetically active layer (Figures 4b and 5) because of the very low nutrient concentrations at the surface and dispersal of light down to a layer in which nutrients are available (Beckmann and Hense, 2007). Photoacclimation also often allows Chl-a to increase in the low PAR layer (Cornec et al. 2021; Masuda et al. 2021). When nutrients are available at shallow depths, Chl-a displays maximum concentrations above the base of the photosynthetically active layer, as in the tropics and subarctic (Figures 4 and 5). Note that the Chl-a concentrations are still significant below the maximum, including those below the photosynthetically active layer in the tropics and subarctic (Figure 3).

The subsurface Chl-a maxima in the global oceans can be categorized into three types depending on their seasonal cycles: (A) a stable maximum below the mixed layer all year round in the subtropics and tropics (30°S to 30°N); (B) a maximum below the mixed layer in summer and within the mixed layer during winter in the midlatitudes (30° to 40°); and (C) a maximum within the mixed layer at 45–60°S in the Southern Ocean and the northern North Atlantic (north of 45°N) (Figures

9 and 10). Thees categories also correspond to the seasonal cycle of surface Chl-a, which is subject to nutrient limitations and is low in all seasons in region A (Figures 10a, 10b, and 10d–f). In region B, nutrients are supplied by winter mixing, and

winter blooms occur in the mixed layer; in contrast, surface nutrients get depleted, and the main body of Chl-a retains in the subsurface during summer (Figure 10c). In region C, Chl-a concentrations in the mixed layer increase after shallowing of the mixed layer when there is sufficient light in summer (Figures 10h and 10i). These features are consistent with regional studies reported in the literature (e.g., Baldry et al., 2020; Fujiki et al., 2020; Mignot et al. 2014; Sverdrup, 1953). The latitudinal dependence of the subsurface Chl-a maxima occurrence was also noted by Cornec et al. (2021).  The low surface

Chl-a concentrations and subsurface Chl-a maximum below the mixed layer in midsummer in the subarctic North Pacific (Figure 10g) are due to summer iron limitation at the surface (Martin and Fitzwater, 1988; Nishioka and Obata, 2017; Nishioka et al., 2020). South of 60°S in the western Indian sector of the Southern Ocean, the subsurface Chl-a maximum is deeper than the mixed layer (Figure 4a), which is consistent with the subsurface Chl-a maximum following sea ice retreat reported by Gomi et al. (2007).

Data on subsurface fluorescence maxima have been sometimes reported without subsurface Chl-a maxima (Falkowski and Kolber, 1995; Biermann et al., 2015). Especially data from profiling floats are potentially suffered from fluorescence quenching at surface (Xing et al., 2012). To investigate subsurface fluorescence maximum, we examined the subsurface Chl-a maximum from each data source. The area-averaged Chl-a concentrations and the subsurface Chl-a maximum show similar seasonal cycles in data from both bottle samples and from profiling floats (Figures S3 and S4). In the Southern Ocean, the

subarctic North Atlantic, and the subarctic North Pacific, the subsurface Chl-a maximum within the mixed layer in winter can be detected in the bottle samples, although the depth of the Chl-a maximum tends to slightly shallower in data from bottle samples than in those from profiling floats (Figures S3g–i and S4g–i). Therefore, a subsurface maximum within the mixed layer is not necessarily just the fluorescence maximum but the substantial Chl-a maximum. This indicates that the subsurface Chl-a maximum is a general feature of the ocean, even in areas with a deep mixed layer in winter. In fact, a

sporadic stratification and the Chl-a maximum just below the sporadic mixed layer have been found in the mid-latitudes and the subarctic in winter (Chiswell 2011; Ito et al. 2015; Matsumoto et al. 2021).

In the subtropics, nitrate concentrations were quite low even at the Chl-a maximum depths (Figure 6c). Nitrate concentration is not necessarily an appropriate index of available nutrients for phytoplankton. Nitrate is used as soon as it is available in the subtropics (Lewis et al., 1986). Furthermore, biological production does not always require nitrate, and ammonium

assimilation is more important in the subtropics than in the subarctic (Eppley and Peterson, 1979). Nitrogen fixation also contributes to biological production in the subtropics (Deutsch et al., 2007; Karl et al., 1997). Meanwhile, nitrate concentrations are high at the depths of the Chl-a maximum in the subarctic North Pacific and in the eastern tropical Pacific because the limiting nutrient for biological production is iron rather than nitrate (Landry et al., 1997; Martin and Fitzwater, 1988; Martin et al., 1990).

Oxygen oversaturation in the subtropics occurs below the mixed layer depth and above the subsurface Chl-a maximum (Figures 2d, 4c, and 5). Biological production per unit Chl-a is generally more effective under high light levels (Yoder,

1979). Oxygen generated in deeper layers would be removed by the remineralisation of sinking particles (Martin et al., 1987). In any case, substantial new production occurs in the subtropics. Seasonal oxygen production in the subtropical subsurface layer from winter to summer sometimes yields approximately 5–10 μmol/kg of oxygen (Figures 7f, 8c, and 8f). When the

seasonal oxygen production is integrated over the 50–150-m depth range, the value becomes 500–1000 mmolO/m$^2$/6 months. Assuming a Redfield ratio of 276O:106C, this is the equivalent of 200–400 mmolC/m$^2$/6 months. This is consistent with the net community production values of $1.6 \pm 0.2$ molC/m$^2$/yr and $0.9 \pm 0.4$ molC/m$^2$/yr reported by Riser and Johnson (2008) in the subtropical North and South Pacific, half of which occurred in the surface layer. The satellite-derived NPP in the subtropics is approximately 20 mmol/m$^2$/day and 3600 mmolC/m$^2$/6 months (not shown here). Consequently, new

production derived from subsurface oxygen production in the subtropics is estimated at approximately 10% of the NPP. This is consistent with a f-ratio of 15% reported at station ALOHA (23°N 158°W; Karl et al., 1996), considering a small but nonzero seasonal dissolved inorganic carbon drawdown reported in the subtropical surface layer (Yasunaka et al., 2013, 2021). It should be noted that oxygen production is not always associated an increase in biomass at shallow depths in the subtropics (Fujiki et al., 2020).

A seasonal reduction in nitrate along with oxygen production in the subsurface layers cannot be inferred (not shown here), because the number and quality of nitrate observations are likely insufficient to detect such a relationship. Another possibility is that nitrogen fixation substantially contributes to biological production (Deutsch et al. 2007; Karl et al., 1997). Lower Chl-a concentrations at the surface during El Niño (Figure 12a) result from reduced upwelling of nutrient-rich subsurface water to the surface (Chavez et al., 1999). Although Matsumoto and Furuya (2011) showed that no substantial

changes in the average subsurface Chl-a concentrations in the western Pacific warm pool region were associated with ENSO, subsurface Chl-a concentrations at the fixed grids were observed here to decrease during El Niño west of 160°E (Figure 11c). The inverse correlation of the surface and subsurface Chl-a concentrations in the central and eastern tropical Pacific associated with ENSO (Figures 11c and 12a) may result from a decrease in surface Chl-a concentrations that increases light penetration to the subsurface (Figures 12a and 12c). This then increases the subsurface Chl-a concentrations, as postulated

by Uitz et al. (2006). In Uitz et al. (2006), the Chl-a profile in stratified water is parameterized to increase the subsurface Chl-a concentrations with decreasing surface Chl-a concentrations. An inverse correlation associated with ENSO has been presented in previous model results (Lee et al., 2014; Kang et al., 2017). However, the subsurface signals in those models are much weaker than the surface signals, unlike those in this study (see Figure 4 in Uitz et al., 2006, and Figure 11 in Lee et al., 2014). Thus, the subsurface response to this process may have been underestimated.

**5 Conclusion**

This study presents the first view of the global maps of subsurface Chl-a maxima and their seasonal variation and interannual variation associated with ENSO. Using in situ Chl-a concentration data, we found dynamic variability in subsurface Chl-a concentrations in time and space. A subsurface Chl-a maximum was observed across the world's oceans, including a stable maximum at depths greater than 80 m all year round in the subtropics and tropics, a maximum below the mixed layer during

summer and within the mixed layer during winter in the midlatitudes, and a maximum within the mixed layer at 45–60°S in the Southern Ocean and the northern North Atlantic. It extends deeper than the base of the photosynthetically active layer in the subtropics but is shallower in the tropics and midlatitudes. At 20–40° latitudinal bands, the subsurface Chl-a maxima tend to deepen in summer with the seasonal deepening of the photosynthetically active layer. The seasonal oxygen increase below the mixed layer in the subtropics implies substantial new biological production. During El Niño, the subsurface Chl-a concentrations in the equatorial Pacific are higher in the middle and to the east and lower in the west than during La Niña, which is opposite to the patterns that occur at the surface.

Chl-a concentrations vary dynamically in time and space on the ocean surface as well as in the subsurface. The maps presented in this study can be used to help validate ocean biogeochemical and Earth system models and should facilitate development of models. Chl-a concentrations are also related to the absorption of shortwave radiation, and the vertical distribution of shortwave radiation affects the thermal structure and dynamics of the ocean (Lewis et al., 1990; Siegel et al., 1995). Therefore, continuous measurements and archiving Chl-a data are desirable. An increase in the coverage of biogeochemical floats with Chl-a sensors is a promising way to generate more subsurface Chl-a data (Chai et al., 2020). This would help reveal the direct relationship in the subsurface layer of the world's oceans between Chl-a concentrations and absolute light intensity, rather than the estimated light intensity described here, and between Chl-a concentration and biomass.

**Data availability**

All data used in this paper are available on the website referred to in the text. Data on the depth of the subsurface Chl-a maxima presented in Figure 2a of this paper is available online (http://caos.sakura.ne.jp/sao/scm/).

**Author contributions**

SY designed the study, conducted the analysis, and wrote the manuscript. TO conceived the study and modified the manuscript. KS provided advice on the analysis and the manuscript. KS helped with data management.

**Competing interests**

The authors declare that they have no conflict of interest.

**Acknowledgments**

The authors thank the two anonymous reviewers for their fruitful comments, and E. Siswanto for his assistance to calculate subsurface PAR.  This work was financially supported by JSPS KAKENHI (Grant Number JP18H04129).

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

**Table 1. Number of Chlorophyll *a* (Chl-*a*) measurements and profiles from each data source with percentages eliminated by quality control.**


| Platform | Num. of measurements | Num. of profiles | Eliminated % |
|---|---|---|---|
| Bottle sample | 1,292,565 | 179,277 | 2.2% |
| CTD fluorescence | 23,824,711 | 89,541 | 0.9% |
| UCTD fluorescence | 1,086,359 | 19,251 | 3.0% |
| Profiling float | 11,383,247 | 67,903 | 0.3% |
| Glider | 75,724,059 | 375,421 | 0.3% |
| Drifting buoy | 796,220 | 6,076 | 0.3% |
| Total | 114,107,161 | 737,469 | 1.1% |

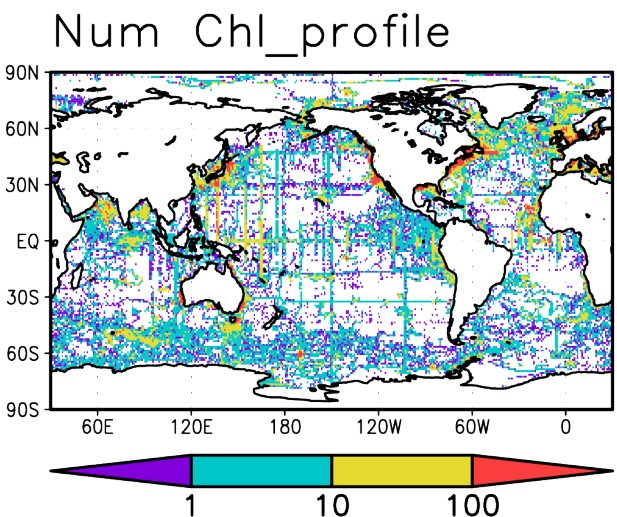


**Figure 1. Numbers of Chl-a concentration profiles.**

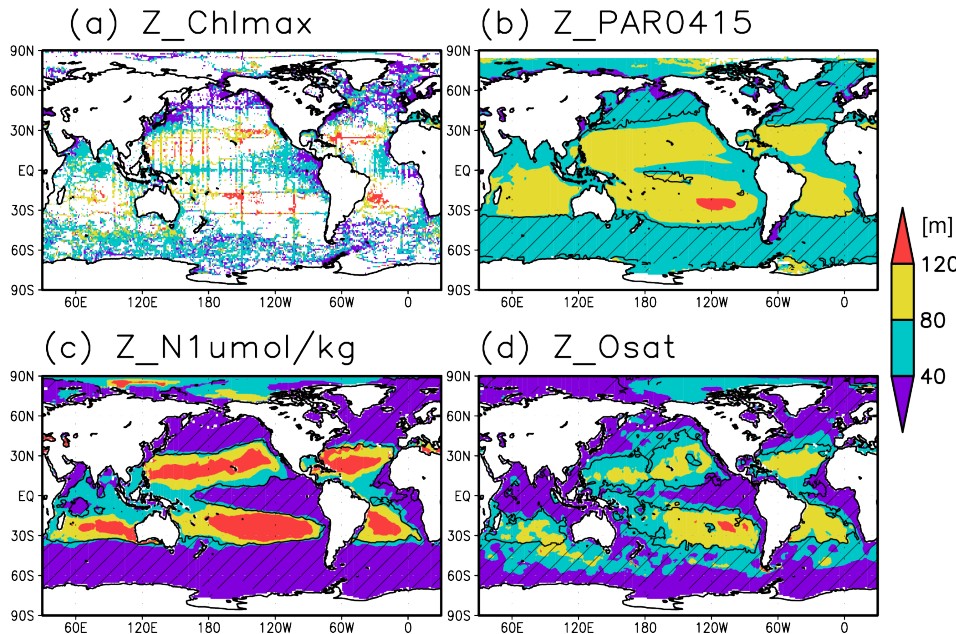

**Figure 2: Depths of (a) the chlorophyll a (Chl-a) maximum, (b) the photosynthetically active layer (>0.415 einstein/m$^2$/day of photosynthetically available radiation), (c) the nitrate depleted layer (<1 μmol/kg of nitrate), and (d) the oxygen-oversaturated layer. The hatched areas show regions in which the mixed layer is deeper than the photosynthetically active layer in panel (b), nitrate-depleted layer in panel (c), and the oxygen saturation layer in panel (d).**


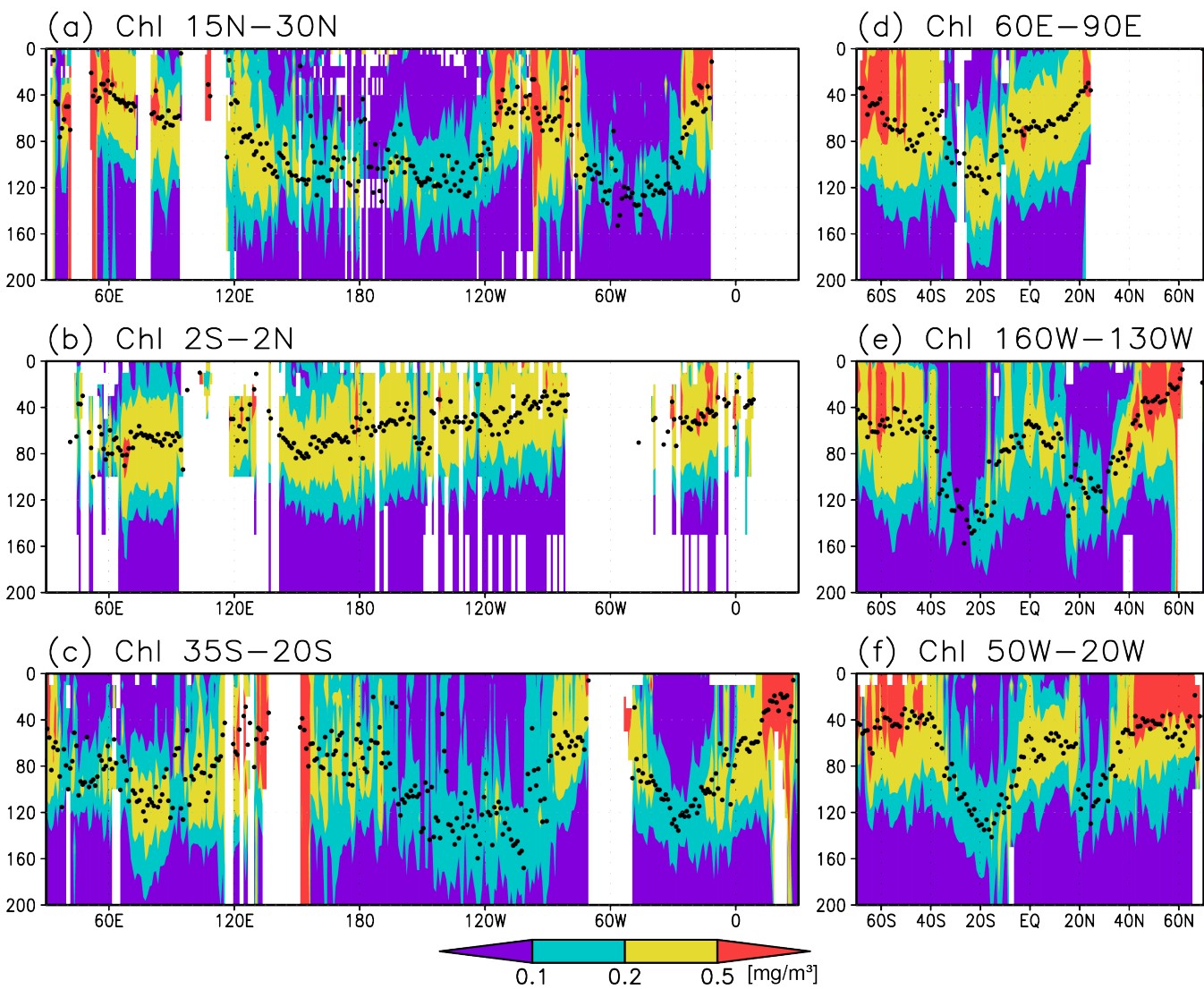

Figure 3. Cross-sections of chlorophyll a (Chl-a) concentrations at (a) 15–30°N (northern subtropics), (b) 2°S to 2°N (tropics), (c) 35–20°S (southern subtropics), (d) 60–90°E (the Indian Ocean), (e) 160–130°W (the Pacific Ocean), and (f) 50–20°W (the Atlantic Ocean). The black dots indicate the depth of the Chl-a maximum.

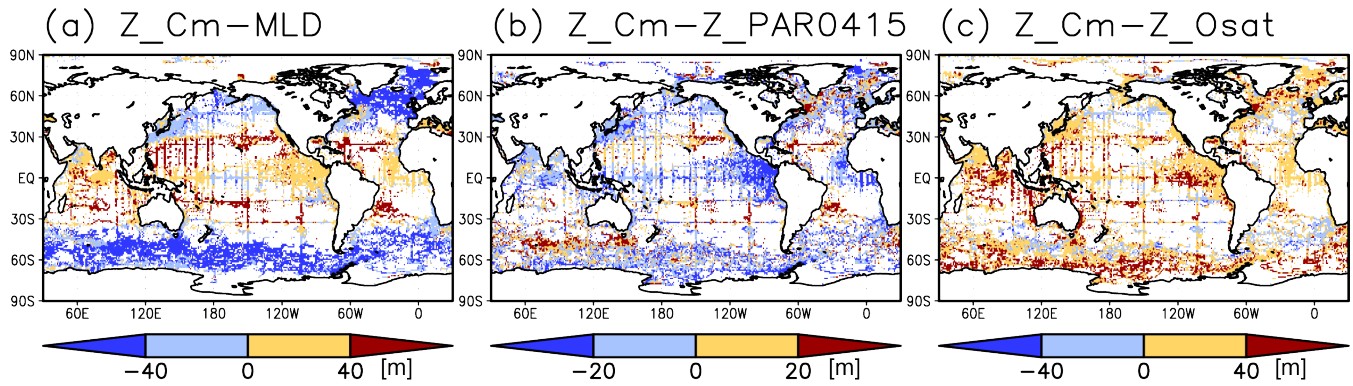

**Figure 4. Depth differences between the chlorophyll a (Chl-a) maximum and (a) the mixed layer, (b) the photosynthetically active layer ($>0.415$ einstein/m$^2$/day of photosynthetically available radiation), and (c) the oxygen-oversaturated layer.**


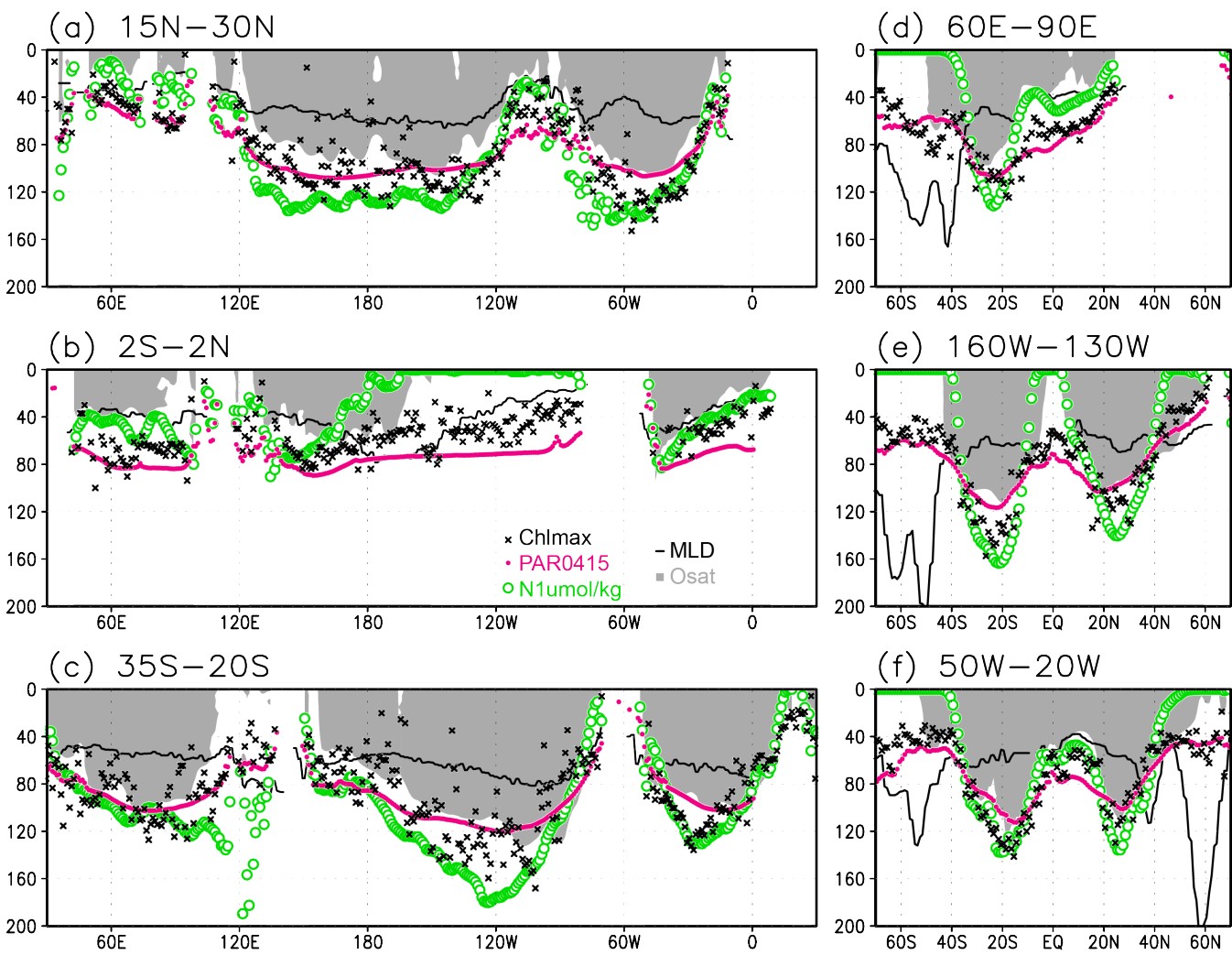

**Figure 5. Depths of the chlorophyll a (Chl-a) maximum (black crosses), photosynthetically active layer (>0.415 einstein/m²/day of photosynthetically available radiation; magenta dots), nitrate depleted layer (<1 μmol/kg of nitrate; green open circles), mixed layer (black line), and oxygen-oversaturated layer (gray shading) at (a) 15–30°N (northern subtropics), (b) 2°S to 2°N (tropics), (c) 35–20°S (southern subtropics), (d) 60–90°E (Indian Ocean), (e) 160–130°W (Pacific Ocean), and (f) 50–20°W (Atlantic Ocean).**


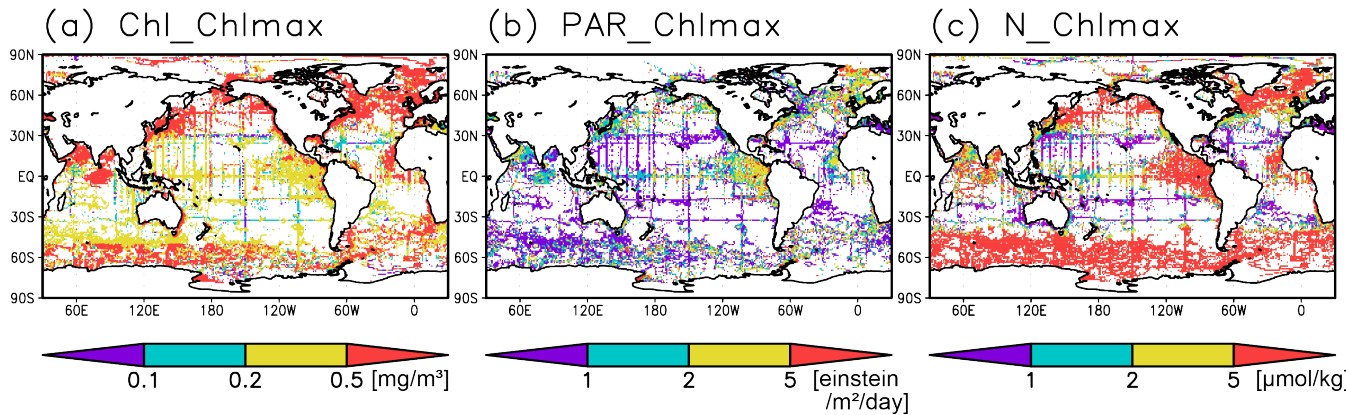

**Figure 6: (a) Chlorophyll a (Chl-a) concentrations, (b) photosynthetically available radiation, and (c) nitrate concentrations at the**
**Chl-a maximum depth.**

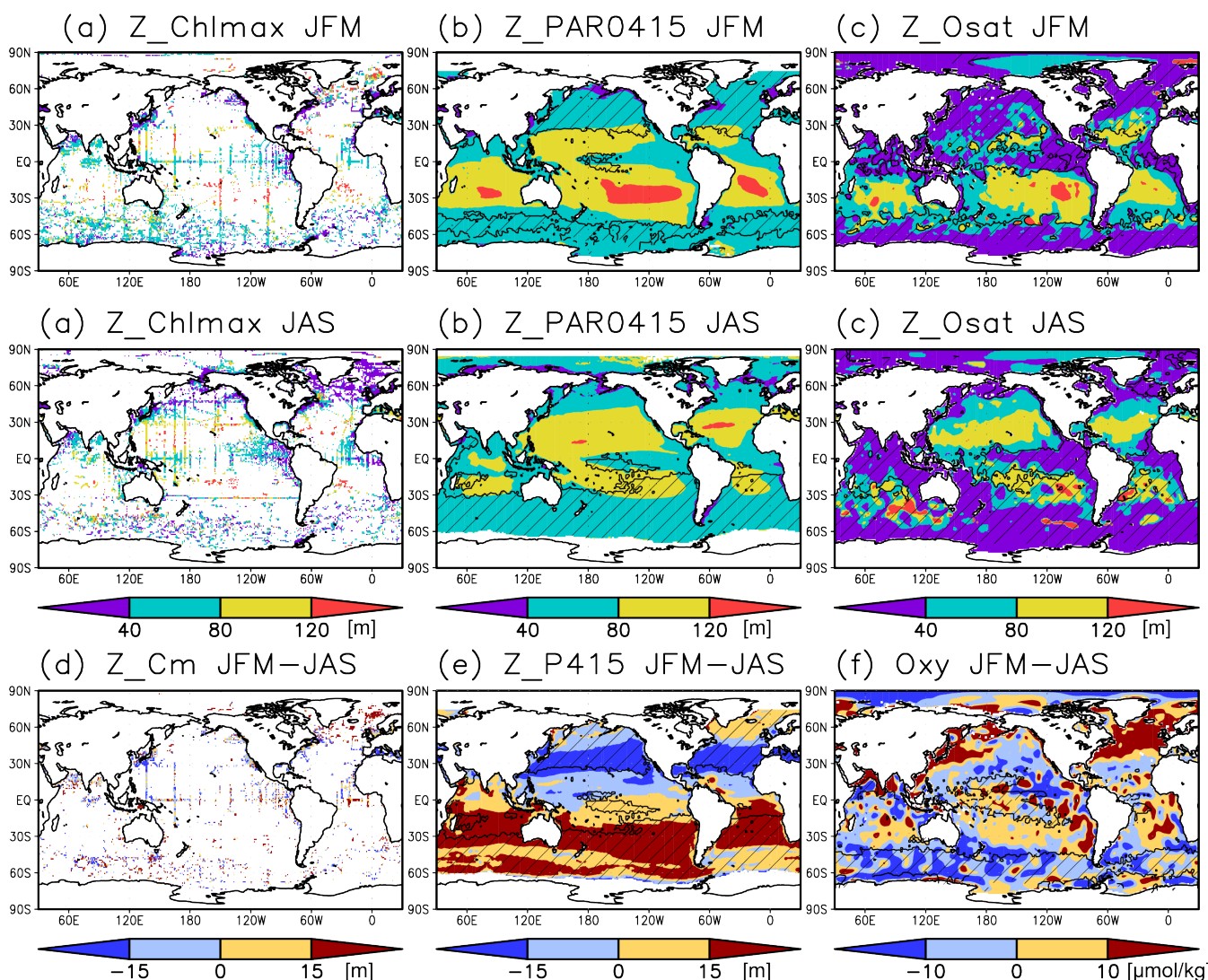

**Figure 7. (a–c) Depths of the chlorophyll a (Chl-a) maximum, photosynthetically active layer (>0.415 einstein/m$^2$/day of photosynthetically available radiation), and oxygen-oversaturated layer in January–March (JFM) (top) and July–September (JAS) (middle), respectively. (d–f) Differences between the Chl-a maximum depth, photosynthetically active layer depth, and dissolved oxygen concentrations at 50–150 m during JFM compared to those in JAS. The hatched areas indicate a mixed layer deeper than the photosynthetically active layer in panel (b), the oxygen saturation layer in panel (c), the photosynthetically active layer in winter in panel (e), and 50 m in summer in panel (f).**


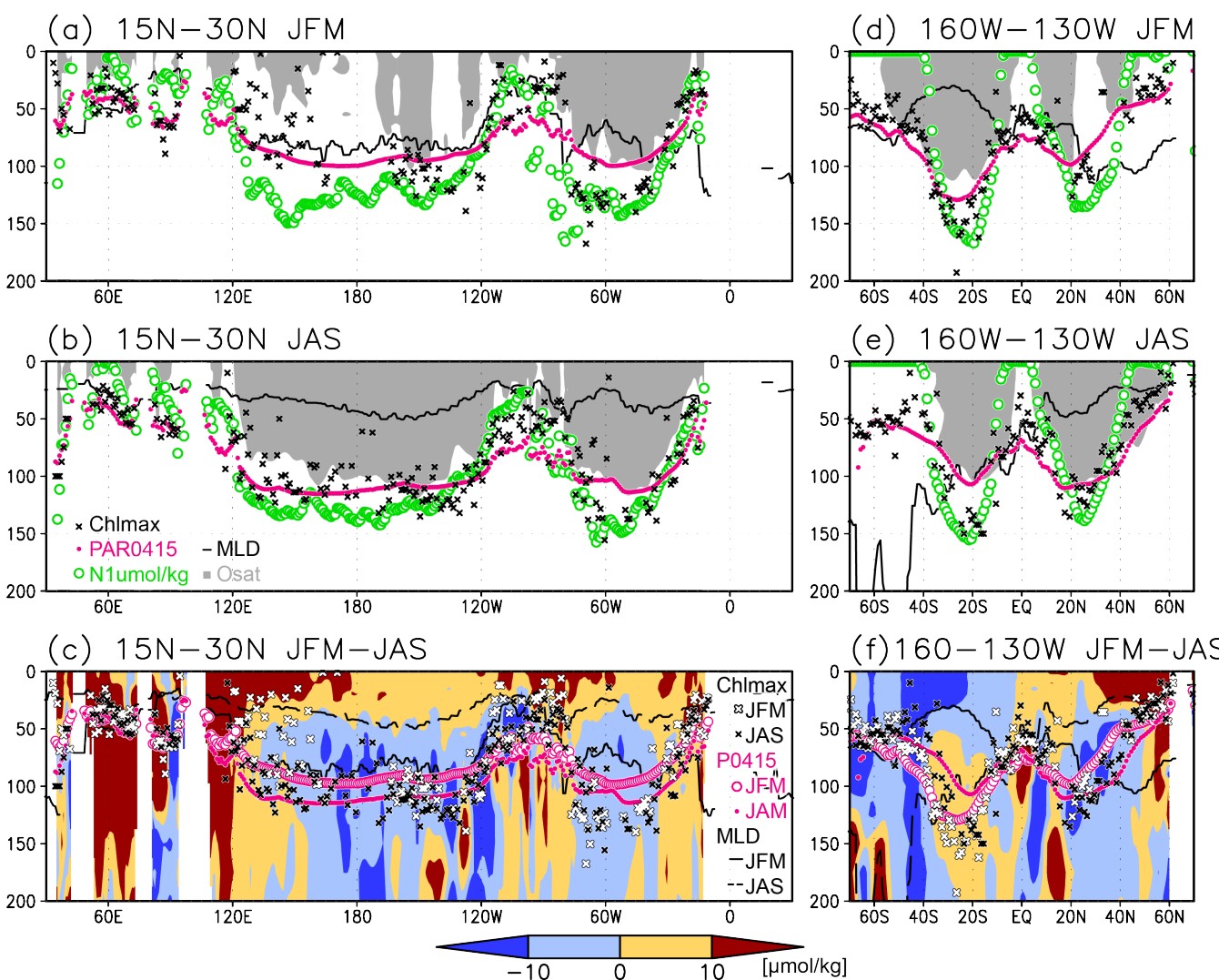

**Figure 8. (a, b) Depths of the chlorophyll a (Chl-a) maximum (black crosses), photosynthetically active layer (>0.415 einstein/m²/day of photosynthetically available radiation; magenta dots), nitrate depleted layer (<1 µmol/kg of nitrate; green open circles), mixed layer (black line), and oxygen oversaturated layer (gray shading) at 15–30°N (northern subtropics) in (a) January– March (JFM) and (b) July–September (JAS). (c) Difference between dissolved oxygen concentrations during JFM and JAS (color scale), and depths of the Chl-a maximum, photosynthetically active layer, and mixed layer in JFM and JAS (white and black crosses, magenta open and solid dots, and solid and dashed black lines, respectively) at 15–30°N. (d–f) Same as (a–c), but at 160– 130°W (Pacific Ocean).**

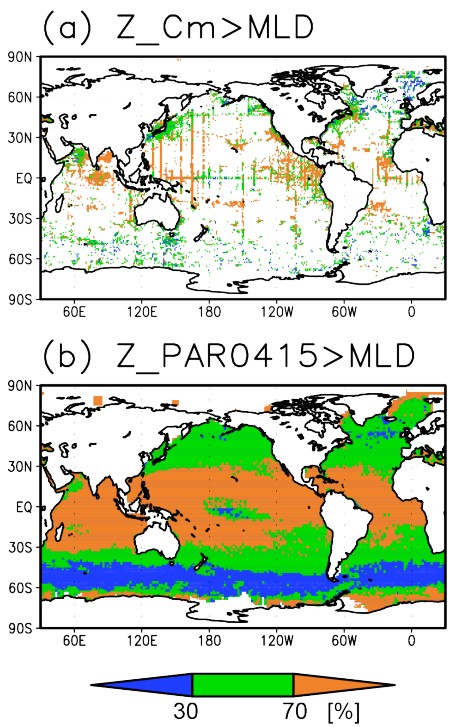

**Figure 9.** Percentage of months when (a) the chlorophyll a (Chl-a) maximum and (b) the photosynthetically active layer (>0.415 einstein/m$^2$/day of photosynthetically available radiation) were deeper than the mixed layer. Data in grids with only one datum were omitted

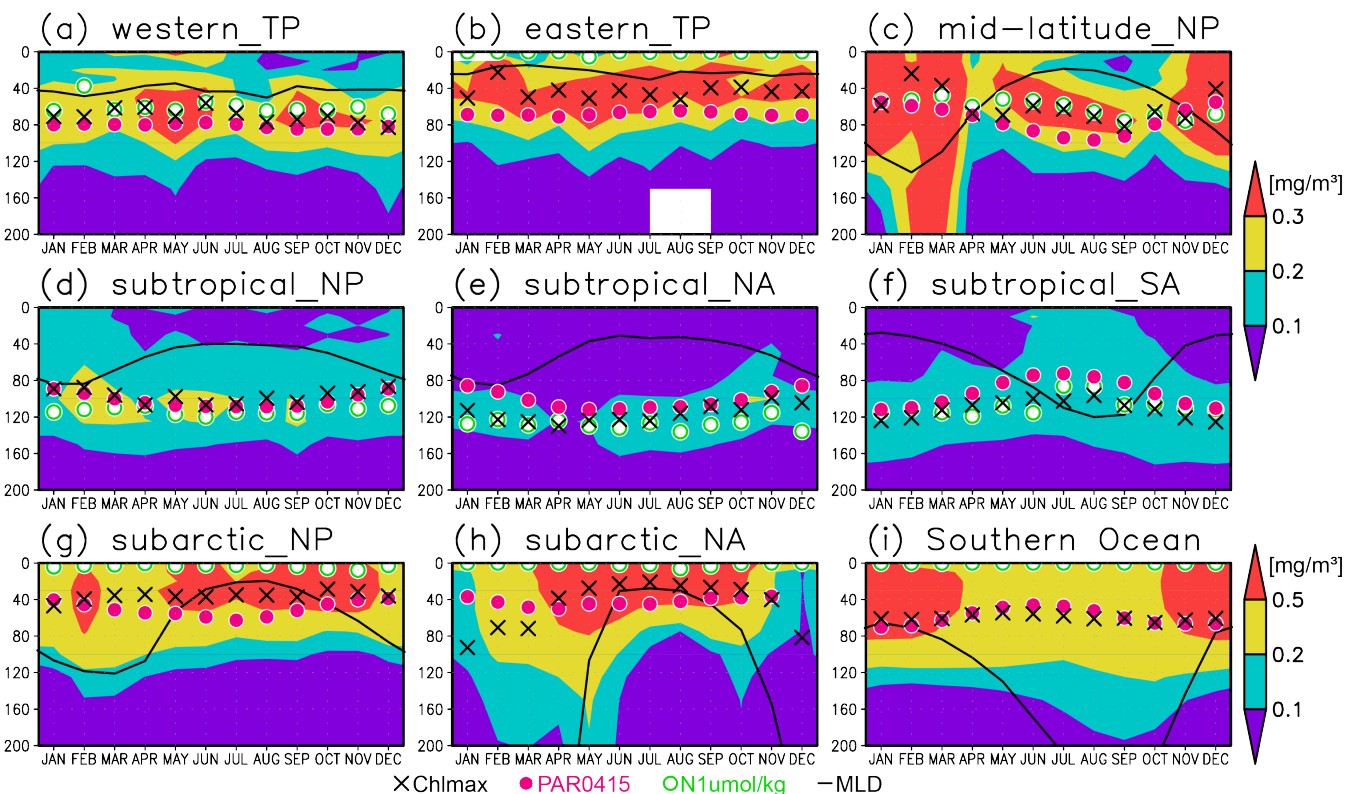

**Figure 10.** Seasonal evolution of chlorophyll a (Chl-a) concentrations with the Chl-a maximum (black crosses), 0.415 einstein/m$^2$/day of photosynthetically available radiation (magenta dots), 1 µmol/kg of nitrate (green open circles), and mixed layer (black solid line) at (a) 2°S to 2°N, 120–170°E (western tropical Pacific), (b) 2°S to 2°N, 120–90°W (eastern tropical Pacific), (c) 30–40°N, 150-130°W (midlatitude North Pacific), (d) 10–30°N, 120°E to 120°W (subtropical North Pacific), (e) 15–30°N, 70–30°W (subtropical North Atlantic), (f) 30–10°S, 50–0°W (subtropical South Atlantic), (g) 40–55°N, 160°E to 155°W (subarctic North Pacific), (h) 50–70°N, 30–0°W (subarctic North Atlantic), and (i) 60–45°S (Southern Ocean).



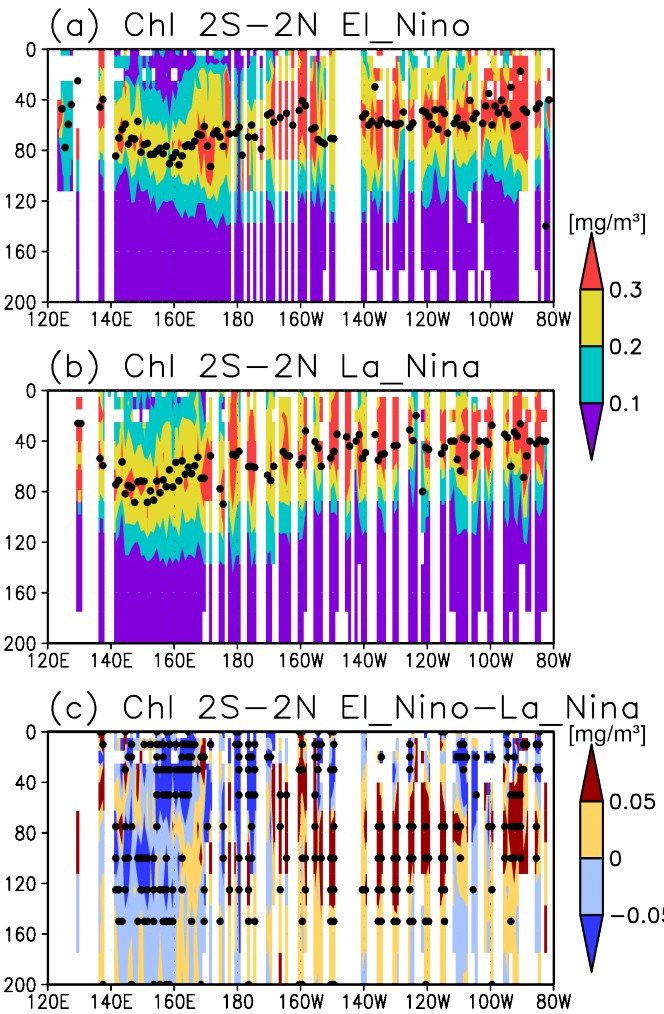

**Figure 11. (a–c) Cross-sections of the chlorophyll a (Chl-a) concentrations in the equatorial Pacific (2°S to 2°N) during El Niño and La Niña, and the difference between them. The black dots denote the depth of the Chl-a maximum in panels (a) and (b), and the significant difference at 5% in panel (c).**


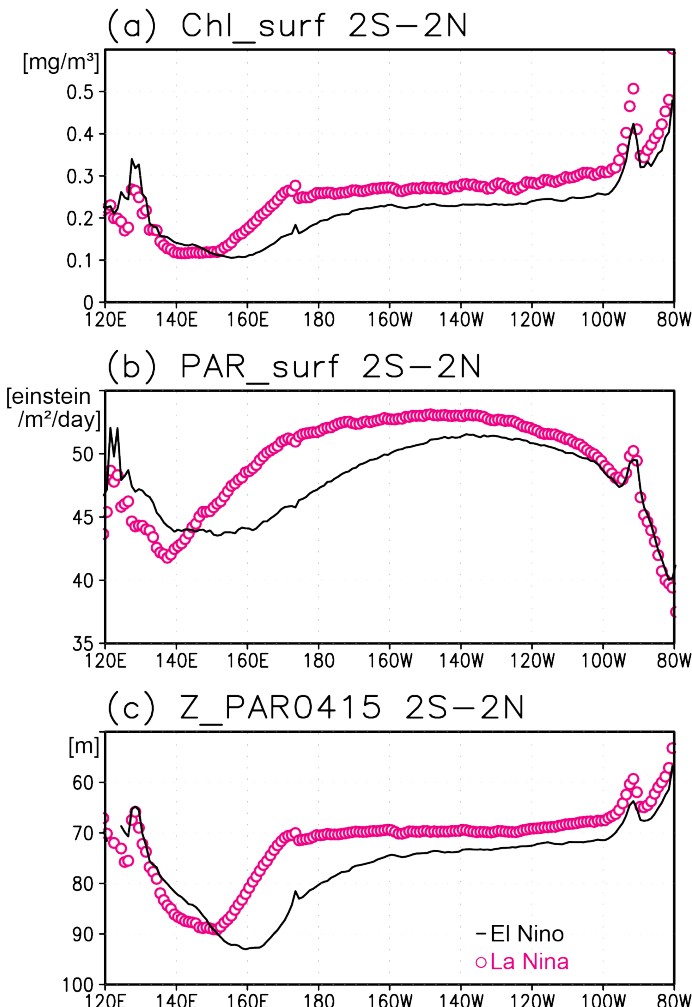

**Figure 12. (a) Satellite-derived surface chlorophyll a concentrations, (b) surface photosynthetically available radiation (PAR), and (c) photosynthetically active layer (>0.415 einstein/m$^2$/day of PAR) along the equatorial Pacific during El Niño (black solid lines) and La Niña (magenta open circles).**