# Peer review of "Global distribution and variability of subsurface chlorophyll a concentrations"

_Ocean Science, 2021_

## Author Comment (AC1)

Reviewer 1 (Dr. Emmanuel Boss):

This paper is focused on describing the salient features associated with the ubiquitous chlorophyll sub-surface maxima in the world's ocean. The depth of this maxima is contrasted with a variety of other horizons (associated with physical stratification, light, nutrient, and oxygen). Both annual and seasonal variation are provided.

This paper is of interest to the readers of OS (as well as Biogeosciences) and I support its publication. While relatively clear, I think this paper could benefit from editing by a native English speaker as some sentences do not read well. I have some comments that I believe if answered could significantly improve the impact of this paper.

We appreciate the acceptance of the potential value of the paper and many constructive comments from the reviewer. Referring to all the comments, we carefully revised the manuscript. The revised manuscript was checked by an English editing service prior to the submission. Point-by-point responses to the reviewer's comments are given below.

Introduction: The introduction mostly avoids discussing the link between Chl-a and phytoplankton biomass. Sometimes the sub-surface chlorophyll maxima is a biomass maxima but often it is mostly a photo-acclimation response of phytoplankton (e.g. Kiefer and Zaneveld, 1991, JGR), e.g. an increase of pigmentation with decreasing light, just as you see with your house plants. This is important to note as many still use Chl-a as an indicator of biomass without understanding the associated bias, particularly when it comes to vertical distribution.

As the reviewer mentioned, the subsurface chlorophyll maxima is sometimes a biomass maxima but it is often a photo-acclimation response of phytoplankton. We have added the notes (Lines 31-32).

Methods: Chlorophyll concentrations are NOT normally distributed in the surface and likely at depth. Thus, it would be good if before applying any QC steps the authors look at the frequency distribution of the data they downloaded and use non-parametric statistics (median, percentiles) rather than averages and standard deviation for QC.

The data types used each has its own source of error (varying from non-photochemical quenching to changes in Fchl/Chl with depth and differences between HPLC and fluormetrically determined Chl). It would have been good if those were detailed. Some

of the uncertainties may have consequences on the profile of Chl. One can see this issue with some of the transects in Fig. 1e where it seems that methodological issues cause a specific cruise line to be different than most other measurements in a given region.

As the reviewer mentioned, the Chl-a concentrations are not normally distributed. Furthermore, erroneous and/or extreme data mostly show quite high values. Although errors of data would be different among the data sources, the statistical quality control using limited number of data did not work effectively, so we treated data together from all data source. Our quality control identified approximately 1% of the measurements as belonging to erroneous or extreme profiles. Data with high values were extensively eliminated, and 85 % of the eliminated data have > 3 mg/m$^3$. The eliminated data were mostly distributed in coastal regions and partly scattered in the open oceans. The ratio o eliminated ratio is slightly larger in data from underway CTD fluorescence and bottle samples probably because they included more uncalibrated data and historical data.

We have added the description of details of the Chl-a data (Section 2); Number of measurements and profiles from each data source (Table 1; Figure S1), distribution of concentration at each depth (Line 61; Figure S2a), and features of the eliminated data by the quality control (Lines 80-84; Table 1; Figure S2b).

Euphotic layer depth, e.g. 1% light level, is a bad light horizon to compare Chl distribution to as phytoplankton care about absolute light levels, not relative ones (e.g. Letelier et la., 2004). In a ML, cells cycling up and down are exposed to, on average, the median light level in that layer. A fixed depth horizon is meaningless in such a case. Also, satellite products of Z_eu do not take into account the vertical chlorophyll distribution but are only based on surface estimate and statistical correlations between those and z_eu, similar to the vertical profiles of Uitz or Westberry. I strongly urge the authors to not use the z_eu but actually attempt to compute the actual light level at the Chl maxima depth (e.g. using a validated model such as Xing and Boss, 2021 that requires PAR at the surface (available from satellite) and a chlorophyll profile).

Euphotic layer depth does not indicate the absolute light intensity as the reviewer mentioned. At the depth of the euphotic layer, the absolute light intensity is larger in subtropics than in subarctic because of the surface PAR is stronger in subtropics. To investigate the absolute light intensity at the Chl-a maximum, we estimated the PAR within the water column from the 1% light depth and the surface PAR as Ito et al.

(2015). The estimated PAR at the Chl-a maximum is mostly stronger than 1 einstein/m²/day in the subarctic and the tropics. We also compared depth of the Chl-a maximum with the photosynthetically active layer (>0.415 einstein/m2/day of PAR; Boss and Behrenfeld 2010). The Chl-a maximum is deeper than the phorosyntherically active layer in subtropics, while it is shallower in subarctic and tropics. That is, the Chl-a maximum is in lower light levels in subtropics than in subarctic and tropics.

PAR within the water column used here is calculated by using surface PAR and 1% light level. As the 1% light level, we used the satellite-based euphotic layer depth estimated by using the statistical relationship between surface Chl-a concentration and the light intensity within water column derived from the in-situ observations by Morel et al. (2007). Thus, it includes the integrated subsurface information although it does not fully take into account the vertical profile.

Although we tried to calculate the actual light level using the formulation by Xing and Boss (2021), it is not successful probably because the profile of the Chl-a concentration described here is too rough to detect the light intensity at the Chl-a maximum. We remain an investigation of the actual light level and the Chl-a concentration in water column for the future work.

As reviewer mentioned, phytoplankton cycles up and down in the mixed layer, and exposed to the various light level. However, the Chl-a maximum that we mainly focus in this study is in deeper than the mixed layer. Thus the comparison of Chl-a maximum with the photosynthetically active layer depth makes sense.

We have used the phorosyntherically active layer (>0.415 einstein/m2/day of PAR) instead of the euphotic layer (1% light depth), and have carefully rewritten the description about the light level including above in the revised manuscript (Lines 89-98, 127-134, 144-149, 154-156, 165, 171, 180-187, 260-262, 271-274).

Results:

The relationship between a chlorophyll maxima and the ML depth depends on the criterion used to define the ML (and there are many). If the ML is an active mixing layer, there cannot, be a maxima within it. If the criterion is such that it describes a longer time scale formation, gradients can form within it (e.g. Zewada et al., 2004, compared optically defined and physically defined ML depth). These can be due to a variety of processes spanning from lateral restratification to phytoplankton

photo-acclimation, to name a few.

The reviewer is right. The mixed layer we used is not the active mixing layer but the mixed layer that represents the history of mixing, and sporadic stratification would exist within it. Actually, a sporadic stratification and the subsurface Chl-a maximum just below the sporadic mixed layer have been found in mid-latitudes and subarctic in winter (Chiswell 2011; Ito et al. 2015; Matsumoto et al. 2021).

We tried to observe the relation with the active mixing layer by using the mixed layer depth with a change from the surface sigma-θ of 0.03 instead of 0.125. However, it did not work well probably because climatological mean is not adequate to show a sporadic stratification.

We added the description above in the revised manuscript (Lines 106-108, 213-215).

Line 155-160- the explanation provided does not take into account the actual light level and hence is correlative at best.

Please see the answer above. We have carefully rewritten the description about the light level including above in the revised manuscript (Lines 89-98, 127-134, 144-149, 154-156, 165, 171, 180-187, 260-262, 271-274).

Line 175-180- If I am not mistaken, iron is undetactable in the subarctic N. Pacific all year.

There are few iron observations in winter in the subarctic North Pacific. But substantial iron is detected in subsurface water in summer that outcrops to the surface in winter (Nishioka et al. 2020). We have added the words that iron limitation is in summer in the revised manuscript (Lines 200-201).

Line 180: In the discussion explain how and why subsurface fluorescene maximum may differ from Chl-a maximum and how it may affect your results. You should attempt to contrast your distribution based on floats and those based on water sample analysis to see if there are significant differences between them.

Following the reviewer's comments, we checked the maximum from each data source. The area averaged Chl-a concentration and the subsurface Chl-a maximum show similar seasonal cycle both in data from bottle samples and in data from profiling floats potentially suffered from fluorescence quenching at surface. In the Southern Ocean, the

subarctic North Atlantic, and the subarctic North Pacific, the subsurface Chl-a maximum within the mixed layer in winter can be detected even in the bottle samples although the depth of Chl-a maximum is slightly shallower in data from bottle samples than in those from profiling floats. Therefore, a subsurface maximum within the mixed layer depth is not necessarily just the fluorescence maximum but the substantial Chl-a maximum. This indicates that the subsurface Chl-a maximum is a general feature of the ocean even in areas with a deep mixed layer in winter.

We have added the description about the Chl-a maximum within the mixed layer depth (Lines 204-213), and the figure of seasonal evolution of Chl-a concentration using data only from bottle samples and profiling floats as Figure S3 and S4.

Line 19: replace 'suggestive results' with 'has implication to'.

Done (Line 21).

Park's paper:

Line 35: If I understand correctly from Park's paper, Chlorophyll responds faster than SST to the wind change associated with El Nino. The way this sentence is written suggests that chlorophyll change is a necessary condition for the onset of El Nino which it is not. This is likely due to the rapid Chl response to local MLD change while the SST change is mostly due to advection (from west to east) which takes months.

The reviewer is right. We deleted this sentence in the revised manuscript.

---

## Author Comment (AC2)

Reviewer 2:

This manuscript provides the global map of the subsurface chlorophyll-a (Chl-a) maximum based on in-situ data. In previous studies, the global map of the subsurface Chl-a maximum is estimated from surface chlorophyll distribution (Mignot et al., 2014), and therefore the map obtained in the paper is valuable to the oceanographic community. The manuscript also reveals seasonal oxygen increases below the mixed layer and the variation of subsurface chlorophyll-a in the equatorial Pacific associated with ENSO. I think that the paper is acceptable for publication after minor revision.

We appreciate the acceptance of the potential value of the paper and helpful comments from the reviewer. Referring to all the comments, we carefully revised the manuscript. Point-by-point responses to the reviewer's comments are given below.

Line 133-135, Line 180-183
The authors state that the Chl-a maxima are often contaminated with the subsurface fluorescence maximum. To me, this seems like an extreme view. If the vertical distribution of fluorescence observed in a mixed layer does not necessarily correspond to that of Chl-a, the significance of measuring fluorescence data in a mixed layer is greatly reduced. Biermann et al. (2015) referred by authors describe the effect of the correction of the fluorescence data but do not seem to describe the difference between fluorescence data and Chl-a data.

Considering the reviewer's comment, we checked the maximum from each data source. The area averaged Chl-a concentration and the subsurface Chl-a maximum show similar seasonal cycle both in data from bottle samples and in data from profiling floats. In the Southern Ocean, the subarctic North Atlantic, and the subarctic North Pacific, the subsurface Chl-a maximum within the mixed layer depth in winter can be seen even in the bottle samples although the depth of Chl-a maximum is slightly shallower in data from bottle samples than in those from profiling floats. Therefore, the subsurface maximum within the mixed layer depth is not necessarily just the fluorescence maximum but the substantial Chl-a maximum. This indicates that the subsurface Chl-a maximum is a general feature of the ocean even in areas with a deep mixed layer in winter.

We have added the description about the Chl-a maximum within the mixed layer depth

(Lines 204-213), and the figure of seasonal evolution of Chl-a concentration using data only from bottle samples and profiling floats as Figures S3 and S4.

Line 234-236

The authors mention that the maps for subsurface Chl-a would be useful to validate ocean biogeochemical and Earth system models. I agree with this opinion. I recommend that the authors publish the data on the subsurface Chl-a maximum depth (Fig. 1a) at a public data repository such as Figshare.

Following the reviewer's suggestion, we will publish the data on the subsurface Chl-a maximum depth at http://caos.sakura.ne.jp/sao/scm/.

Line 126-127

Does the calculation of euphotic layer depth take into account the light shading effect of phytoplankton? If the shading effect is considered, I can understand the sentence (North of …).

In the revised manuscript, since we used the photosynthetically active layer instead of the euphotic layer, this sentence was deleted.

---

## Author Response (AR2)

Editor (Dr. Piers Chapman):

I have gone through the revised manuscript and feel that the authors have answered all the questions posed by the reviewers. The paper can now be published, but there are a number of minor points that need to be addressed first. Additionally, I believe the copy editors will have some changes to the English.

We appreciate the positive evaluation and the helpful comments from the editor.

1. Line 53: I think this should read "We then present seasonal and interannual…"

We have deleted "clarify" in the revised manuscript (line 54).

2. Line 87: By "sampling depth interval" do you mean "thickness of the chl-a maximum"?

We have changed the description to "sampling depth interval around the Chl-a maximum" in the revised manuscript (line 87).

3. Line 104: I think this would read better as "(the depth at which sigma-theta changes by 0.125 compared to that at the surface)"

Done (line 104).

4. Line 112: "Here, El Niño or La Niña are taken to refer to all positive or negative…"

We have changed the description to "Here, El Niño or La Niña is taken to refer to all positive or negative ... (line 112).

5. Lines 154-156: I still have no idea what this means, nor what Fig. 9 actually shows. Is it important or relevant?

We apologize that we did not noticed the comment from the editor in the previous round of review. We have changed the description to "The percentage of months with a deeper photosynthetically active layer than the mixed layer has a similar pattern to the percentage of months deeper the subsurface Chl-a maximum than the mixed layer" (lines 154-156). We also have changed Figure 9 from "ratio" to "percentage", and the titles of Figure 9 to "Z_Cm>MLD" and "Z_PAR0415>MLD".

6. Line 208: I think there is something missing here. It looks as though a sentence

should end after "and from profiling floats", but then what comes before "potentially suffered from…"?

We have added the sentence "Especially data from profiling floats are potentially suffered from fluorescence quenching at surface (Xing et al., 2012)." in line 206.

References

Kang et al is given as 2017 in the reference list, but as 2018 (line 40) and 2019 (line251). Which is it, or are there more than one references to this author?

Both of them are 2017. We have corrected them in the revised manuscript (lines 40 and 252).

Hosoda et al (line 105) not in reference list

Nishioka et al 2020 (line 201) not in reference list

Chiswell et al (line 214) not in reference list

We have added them in the reference list.